

# Rapid detection and quantification of Nile Red-stained microplastic particles in sediment samples

Masashi Tsuchiya[1,*], Tomo Kitahashi[1,2,*], Yosuke Taira[3], Hitoshi Saito[3], Kazumasa Oguri[1,4], Ryota Nakajima[1], Dhugal J. Lindsay[1] and Katsunori Fujikura[1]

[1] Japan Agency for Marine-Earth Science and Technology (JAMSTEC), Yokosuka, Kanagawa, Japan
[2] KANSO Technos Co., Ltd., Tokyo, Japan
[3] NEC Corporation, Tokyo, Japan
[4] Danish Center for Hadal Research and Nordcee, Department of Biology, University of Southern Denmark, Odense, Denmark
* These authors contributed equally to this work.

Corresponding author
Masashi Tsuchiya,
tsuchiyam@jamstec.go.jp

## ABSTRACT

The distribution and migration processes of microplastics (MPs) in the marine sediments have yet to be fully elucidated. To estimate the contamination levels and distribution patterns, and develop countermeasures, the amount of MPs must be understood. Rapid and efficient processing of numerous samples is also needed to detect and determine MP contamination. However, whatever the sample of interest, MP analysis is time consuming. This is especially the case for deep-sea sediments, where the particle sizes are small and pretreatment processes are complex and time-consuming. To address the need for rapid and efficient detection of MPs, we propose a novel method for automatically identifying and counting Nile Red (NR)-stained sedimentary MP particles captured under a stereoscopic fluorescence microscope. In this study, we demonstrated the utility of the developed system by comparing its recovery rate and analysis time with those of the conventional methods used for manual processing. The developed method can efficiently detect MPs of sizes between 18 and 500 μm and classify them as fibers or grains (or fragments). This means that our method can efficiently detect MPs as small as 100 μm found in deep-sea sediments. The semi-automated MP detection system gave a counting time of 4.2–8.8 s per particle—as the number of particles increases, the analysis time per particle decreases. Similarly, when the number of particles counted using a stereomicroscope and image analysis software was set at 100, the automatic measurement method using a flow cell could measure 50–80% of the total number of particles, depending on the type of MPs. By using artificial particulate and fibrous MPs as training data and combining them with a machine learning system, we were able to build a system that can classify both types with 98% accuracy (100% for fibers and 96% for grains). In natural samples, approximately 150 μm (20–350 μm in range) MPs were detected, and the number was consistent with previous studies. This demonstrates the effectiveness of the method we developed. We established a rapid detection method for the number and form of MPs using a continuous semi-automated method, combining NR staining and artificial intelligence. Although this method does not allow the identification of polymer types, it enables that rapid

and reliable quantification of MPs numbers. The new method established in this study is expected to improve the accuracy of information on the distribution, destination, and quantity of MPs. It is also relatively easy to use and can transfer technology in various fields, from citizen science to rapid diagnosis on research vessels in the open ocean.

## INTRODUCTION

Elucidating the behavior of microplastics (MPs) introduced into the ocean is an important global problem and is an urgent issue that needs to be addressed. Marine MP pollution is widespread not only on the ocean surface (*Jambeck et al., 2015*; *Isobe et al., 2019*), but also in deep-sea sediments in various oceanic regions (*Van Cauwenberghe et al., 2013*; *Bergmann et al., 2017*; *Maes et al., 2017b*; *Tsuchiya et al., 2024*). However, ninety-five percent of the plastics (*Isobe & Iwasaki, 2022*) are "missing" (*Jambeck et al., 2015*; *Lebreton et al., 2017*). Understanding the total amount of MPs in deep seafloor sediments will help us find such "missing" plastics. On the other hand, MPs with sizes <100 μm, which have frequently been observed in deep-sea sediments (*Van Cauwenberghe et al., 2013*; *Bergmann et al., 2017*; *Barrett et al., 2020*; *Courtene-Jones et al., 2020*; *Cunningham et al., 2020*; *Tsuchiya et al., 2024*), are difficult to handle with tweezers. Although the methods used in previous studies such as μ-Fourier-transform infrared spectroscopy (FTIR), can accurately analyze polymer types of these small MPs, they are time-consuming, which has been a bottleneck for the rapid measurement of large numbers of sediment samples.

Actual *in situ* conditions of MPs remain unclear, especially in deep-sea sediments. Although deep-sea sediments are considered as a major sink of MPs (*Woodall et al., 2014*; *Kanhai et al., 2019*; *Näkki, Setälä & Lehtiniemi, 2019*; *Kane et al., 2020*), studies of MP distributions at the deep-sea floor have been limited and the actual form of the distribution and the transportation processes involved in MP supply to the deep-sea remain unclear. Therefore, to examine MP pollution efficiently, innovative counting methods, such as automatic detection and quantification, must be established to develop a rapid analytical method (*Roch & Brinker, 2017*; *Prata et al., 2019*) to clarify the actual amounts, distributions and types of MPs.

Although the fluorescent staining method to detect MPs is not a spectroscopic or chemical analysis and cannot identify the type of the polymer, it is a useful and inexpensive method for detecting MPs rapidly and in large quantities. Fluorescent reagents, mainly Nile Red (NR), are widely used for microplastic staining (*Maes et al., 2017a*; *Chatterjee et al., 2023*); NR was originally used for cell staining in general, especially for lipid staining in cytological studies. It is a fluorescent reagent with an excitation peak at 450–490 nm and a fluorescence peak at 515–565 nm, and it is relatively easy to stain MPs for detection. While the Nile Red staining method has historically had difficulty identifying the type of plastic, recent work has reported on a system that combines multiple optical filters and

machine learning to identify the material of the plastic as well as particles other than plastic (*Meyers et al., 2022*). It has been applied to natural samples containing MPs that are affected by weathering (*Meyers et al., 2024a*) and to the detection of particles incorporated into organisms (*Meyers et al., 2024b*). It would also be useful to introduce a simpler method to detect smaller particles and to investigate the abundance and distribution of MPs for the selection of survey sites during a survey cruise in near-real time.

Analytical methods, counting and semi-automatic measurement methods, for NR-stained MPs have been devised for MPs collected on filters (*Erni-Cassola et al., 2017*; *Maes et al., 2017a*; *Prata et al., 2020*). In the method presented in this study, the samples are acquired on video and particle occurrences are extracted from the captured video. By loading the captured video into the system, it is possible to simultaneously discriminate the type (fibrous or particulate), count the number, and measure feret diameter of MPs. This enables the rapid and efficient analysis of large numbers of samples.

We aimed to efficiently detect small MP particles, a time and labor limiting factor in MP analyses, and to collect primary data while providing preliminary data on MP distribution patterns, so we developed this flow cell-based, semi-automated MP particle measurement system. This system is also designed to facilitate efficiency during scientific cruises, such as providing near-real time data for the selection of observation points. To achieve this goal, we developed continuous counting and measurement of NR-stained MP particles under a fluorescence microscope. We evaluated the effectiveness of this method by examining MP particles contained in marine sediment between 18 and 500 μm in size, which were measured continuously under a fluorescence microscope, and methods for their automatic identification and efficient counting were investigated. The proposed analytical method is expected to increase work efficiency and improve the calculation accuracy of the distribution, destination, and total number of MPs in environmental samples. It should help alleviate the bottleneck in the acquisition of data through shifting from manual counting and preparation to an automatic detection method.

## MATERIALS AND METHODS

### Overview

In this study, we developed a semi-automatic MP detection (SAMPdetect) system to count the number, measure the size, and assess the shape of MPs stained with NR through the following: (A) production of a flow cell to detect MPs for efficient pass-through under a fluorescence stereomicroscope, (B) improvement of detection methods using artificially created MPs, and (C) validation of the effectiveness of the system using natural sediment samples. The proposed detection system is comprised of two components (Fig. 1A): (1) a flow system (flow cell) designed so that NR-stained MP particles continuously pass in front of the objective lens of a fluorescent microscope, and (2) an automatic detection system for NR-stained MP particles through machine learning methods (Fig. 1B).

For the detection of MPs, we devised the following steps: (a) staining of MP particles, (b) introduction into the flow cell, (c) acquisition of movies by fluorescent stereomicroscope-mounted camera, (d) particle detection and measurement from the acquired movies, and (e) shape identification of the detected particles.
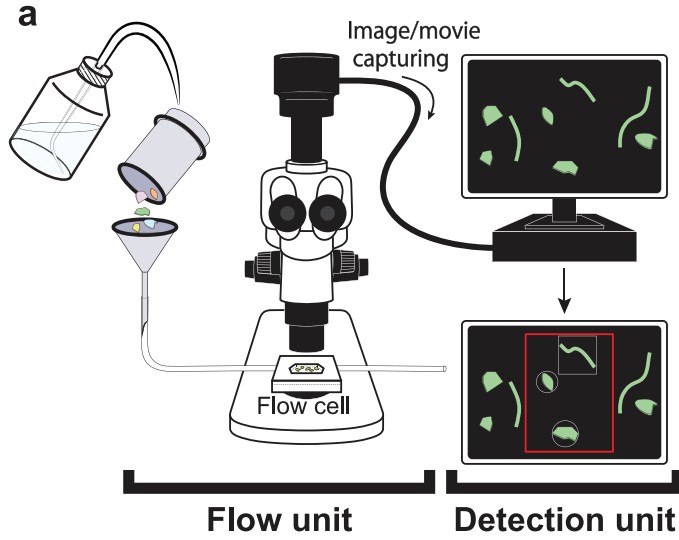

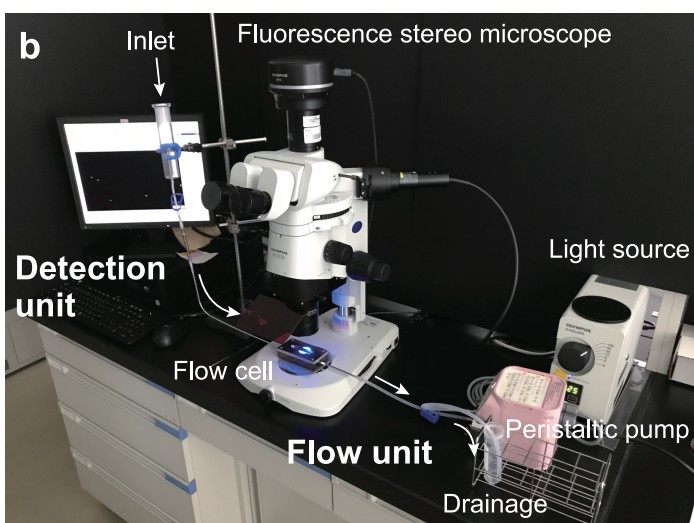

**Figure 1** **Overall configuration of semi-automated microplastic detection (SAMPdetect) system.** (A) Overview and (B) the assembled SAMPdetect system. The flow cell was connected to the introduction and discharge tube to provide a flow path. A glass funnel (glass syringe) was installed in the forefront of the introduction part and the Nile Red-stained MPs were added to the flow cell with water or ethanol *via* the glass funnel. A peristaltic pump was installed in the discharge part to keep the MPs constantly moving by maintaining the flow rate.

## Development of flow cell

The flow cell was comprised of two stainless steel plates (Fig. 2A). To avoid turbulence and clogging of the MPs in the flow path, we made an elongated hexagonal observation chamber. An elongated hexagonal groove with a depth of 0.5 mm, width of 12 mm, and length of 61 mm was cut in the lower plate, and a hole with a diameter of 3 mm was drilled near (approximately 1 mm) both ends. This design was made to connect to an external tube through these holes. On the top plate, a similar sized elongated hexagonal window was made. A cavity, through which NR-stained MPs and liquid solutions were passed, was

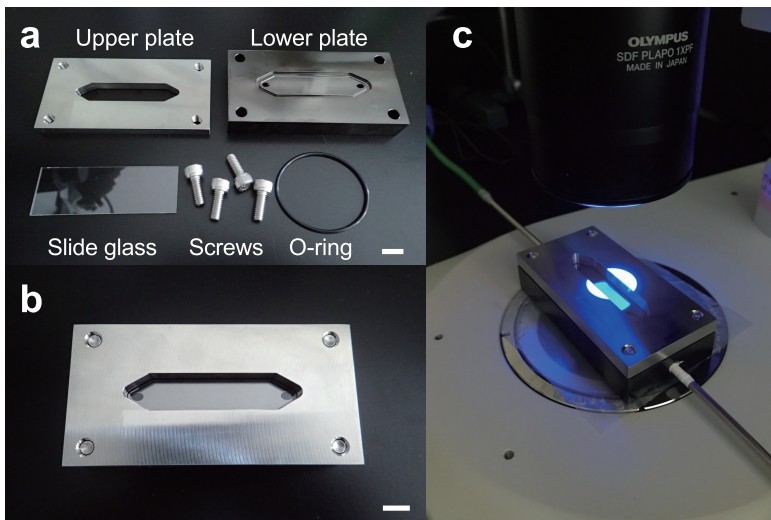

**Figure 2 Configuration and installation of the flow cell.** The flow cell consisted of two stainless steel upper and lower plates. (A) A boat-shaped hexagonal groove with a depth of 0.5 mm, width of 12 mm, and length of 61 mm was cut in the lower plate and connected to an external tube through 3 mm diameter holes at both ends. (B) A boat-shaped hexagonal window was drilled in the top plate and a 76 × 26 mm square glass slide was inserted to create a space for the liquid and microplastics to flow. (C) Example installation of a flow cell under a fluorescence stereomicroscope. Scale bar indicates 1 cm.

created by inserting a 76-width × 26-depth × 1-height mm rectangular glass slide between the two stainless steel plates (Fig. 2B). The lower plate was coated with nickel to prevent irregular reflection of the light source from the fluorescence stereomicroscope. The flow cell was then placed under the fluorescence stereomicroscope (Fig. 2C) and images and video images of NR-stained MP particles were acquired (see below). This system was set to a 10× objective field of view. The field of view of the microscope (depth) was 11.2 mm and covered 93.3% of the flow cell width.

The flow cell was connected to the inlet and discharge tubes to provide a flow path (Figs. 1A and 1B). The inlet tube allowed NR-stained MPs to enter the flow cell, while the discharge tube facilitated the exit of the particles. A glass funnel (glass syringe) was installed upstream of the inlet tube, and NR-stained MPs were added to the flow cell with water or ethanol *via* the glass funnel. A stainless-steel spatula or mixer was used to stir the MPs in the funnel to prevent clumps of particles being introduced into the flow cell. In addition, a peristaltic pump (Smoothflow Pump QT-100; Tacmina Corporation, Osaka, Japan) was installed in the discharge setup to facilitate constant movement of MPs, while maintaining a steady flow rate. We also created an intermittent actuator to control the discharge rate of the peristaltic pump and used it in combination to eliminate particle tailing during image acquisition. A bottle was installed in the discharge setup at the terminus to collect the MPs that passed through the flow cell. The MPs were observed under a microscope and were hand-sorted or collected on a membrane filter for comparison with the automatic detection system.

## Detection of NR-stained microplastics

NR-stained MPs were loaded with water or ethanol into the introduction tube and passed through the flow cell set up under the fluorescent stereomicroscope (SZX16; Olympus Corporation, Tokyo, Japan). This was attached to a fluorescent light source (U-HGLGPS; Olympus Corporation, Tokyo, Japan) and digital camera (DP74; Olympus Corporation, Tokyo, Japan) operated using the cellSens™ software (Olympus Corporation, Tokyo, Japan). When ethanol is used as the solvent, MPs sink due to their low specific gravity and can pass through the flow cell and its connected tubes. MPs were flushed at a flow rate of 2.0 mL per min. We detected NR-stained MPs and imaged them with excitation light of wavelength 450–490 nm and a fluorescence wavelength of 515–565 nm. The movies were acquired at a magnification of 10× with an exposure time of 50 milliseconds (ms) at 100% illumination intensity and a gain of 1×. We set the exposure time to 50 ms because it was found to identify particles best, after multiple tests to acquire movies. This exposure time was sufficient to acquire the clear fluorescent signal from MPs without blurring with the intensity of the excitation light source. The image was trimmed to reflect the pulsation of the peristaltic pump, with the beginning and end synchronized with each of its movements. Seven trials, specifically two measurements of each of the three types of artificial MPs and one natural sediment sample, were conducted. To verify the effectiveness of the automatic detection system, we compared the number of MPs and analysis time of the automatic detection system with those of the manual handling process.

## Detection of NR-stained artificial microplastics

To evaluate the effectiveness of the automatic detection system, we used ground virgin MP particles and microfibers. Three polymer types were ground up to produce the MP particles: polyethylene (PE), polypropylene (PP), and polystyrene (PS). These plastics are commonly made, commercially available, and account for >50% of the total plastic volume produced globally (*Yeo et al., 2018*). In fact, the proportion of these three types of MPs in the natural environment was about 90% in the surface layer of Tokyo Bay (*Nakano, Arakawa & Tokai, 2021*) and 53–95% in deep-sea sediments (*Tsuchiya et al., 2024*). They were processed using a plastic cutting mill (PLC-2M; Osaka Chemical Co., Ltd, Osaka, Japan) to produce MPs with particle sizes of <500 μm, containing various particle sizes and types.

Furthermore, grinding by a mill does not necessarily cut the plastics into particles that fall only within the given size range, with particles both below and above the lower and upper size limits being formed. The size of the artificial MPs falls around the median of the recorded fractionated particle sizes. The artificial MPs had relatively high circularity and a low aspect ratio (Table 1). Fibrous MPs derived from clothing were separated from washing machine drainage.

The artificial MPs were stained with NR at a concentration of 10 μg/mL in hexane (Fujifilm Wako Pure Chemical Corporation, Osaka, Japan) for 1 h and passed through a flow cell. NR-stained artificial MPs were introduced into the flow cell and counted using the automatic detection system before counts were checked under the microscope. We also measured the grain size and general morphology of the MPs (grains or fibers).

**Table 1 Size, circularity, and aspect ratio of collected microplastics.**

| Sample type/number | | Diameter (μm) | | Circularity | | Aspect ratio | |
|---|---|---|---|---|---|---|---|
| | | Avg. | Stdv. | Avg. | Stdv. | Avg. | Stdv. |
| Polyethylene | #1 | 387.9 | 205.4 | 0.48 | 0.16 | 2.59 | 1.16 |
| | #2 | 350.6 | 128.3 | 0.55 | 0.14 | 2.71 | 2.41 |
| Polypropylene | #1 | 292.8 | 157.9 | 0.62 | 0.15 | 2.26 | 1.00 |
| | #2 | 277.5 | 112.5 | 0.64 | 0.14 | 2.07 | 0.84 |
| Polystyrene | #1 | 249.8 | 108.8 | 0.60 | 0.16 | 2.00 | 0.72 |
| | #2 | 243.1 | 120.4 | 0.72 | 0.11 | 1.96 | 0.67 |
| Nojima (natural sediments) | — | 152.9 | 94.1 | 0.42 | 0.20 | 3.16 | 1.51 |

**Note:**
The values measured by cellSens are indicated.

The experimental MPs were filtered and manually counted after passing through the automated detection system (Fig. 3). For comparison, the captured images and the movies that determined the number of MPs and the detection time recorded by the automatic detection system were compared with other processing procedures (1. counting under a microscope by the naked eye (indicated as Microscope), 2. visual measurement from captured images by the naked eye (indicated as picture visual), 3. counting by the naked eye while playing the movie at 2× speed (indicated as Movie visual), 4. measurement by image analysis software (indicated as cellSens), and 5. SAMPdetect method developed in this study (indicated as Movie auto)). Methods 1, 2, and 3 can count the number of MP particles (hatched with grey in Figs. 4A–4D and 5A–5D), and 4 and 5 can simultaneously count the number of MP particles and measure their size. In Methods 1, 2, and 3, we performed counts in triplicate to assess human observational error. In Methods 4 and 5, we automatically measured all the values under the same conditions in the software; hence, requiring only one measurement. These procedures were performed for all three types of the MP materials. By analyzing the different numbers of particles in each run, we investigated correlations between the number of detections and detection time and compared the SAMPdetect method with the other detection methods.

## Detection of NR-stained microplastics in nature

To assess the effectiveness of SAMPdetect method in detecting MPs in natural environments, we used natural sediment samples collected from Nojima Channel, Hirakata Inlet, facing Tokyo Bay (35°19.353′N, 139°38.072′E), Japan, using glass bottles and an aluminum sediment core tube (Tsuchiya et al., 2019). The collected samples (from 1 m water depth) were brought back to the laboratory and analyzed with the automatic detection system developed in this study and described above. A portion of the sediment sample was separated to determine the water content and the dry weight of the sediment used (20.1 g) was calculated. Prior to applying the MPs to the flow cell, sediment samples were separated into larger particles with a 500 μm sieve and then cleaned with a 32 μm sieve. The fine fraction (32–500 μm) was used for the analyses.

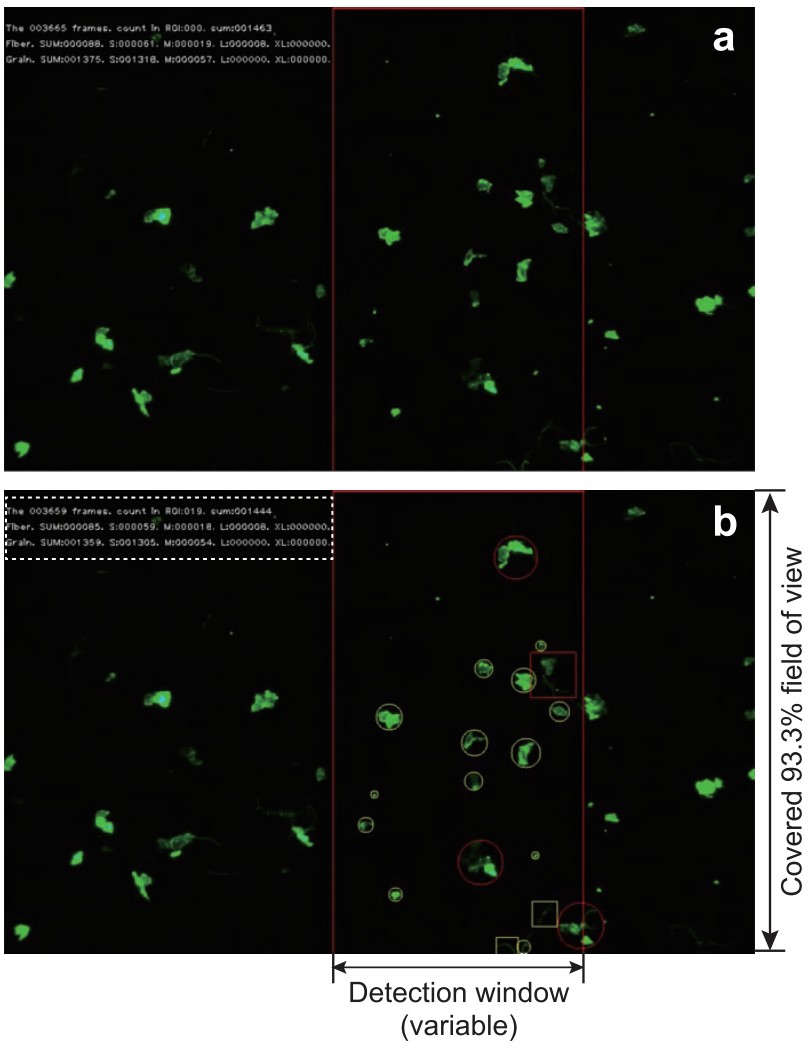

**Figure 3 An example of capturing a video during detection.** The red frame in the screen is the detection window. When (A) particles enter the detection window and (B) the system identifies a grain by marking it with a circle and a fiber by marking it with a square. The maximum diameter of the circle (or the length of the square) is calculated. The frame number, number of counts in the frame, and total number of counts are displayed in the upper left corner. In addition, the number of each size category of fiber or particle is displayed. The frame number, total number of counted MPs, subtotal number of both fibers and grains, subtotal number of each size are shown in the dotted box in (B).

Washed sediments were used and the MPs were separated from the sediment using a JAMSTEC microplastic-sediment separator (*Nakajima et al., 2019b*) by density separation using with an aqueous sodium iodide (NaI) solution (specific gravity of 1.6 g cm$^{-3}$). The density-separated MPs were concentrated in a concentrator using a stainless-steel sieve designed to fit for the size of laboratory beakers (*Nakajima et al., 2019a*). They were further treated with 1N HCl for 30 min in a glass beaker to dissolve the carbonates, thereby preventing their autofluorescence, and remove biogenic calcium carbonate materials such as fragments of seashells, ostracods and foraminiferal tests. Then, Fenton's reagent (*Tagg et al., 2017*) was used for further removal of organic materials from the sample sediment.
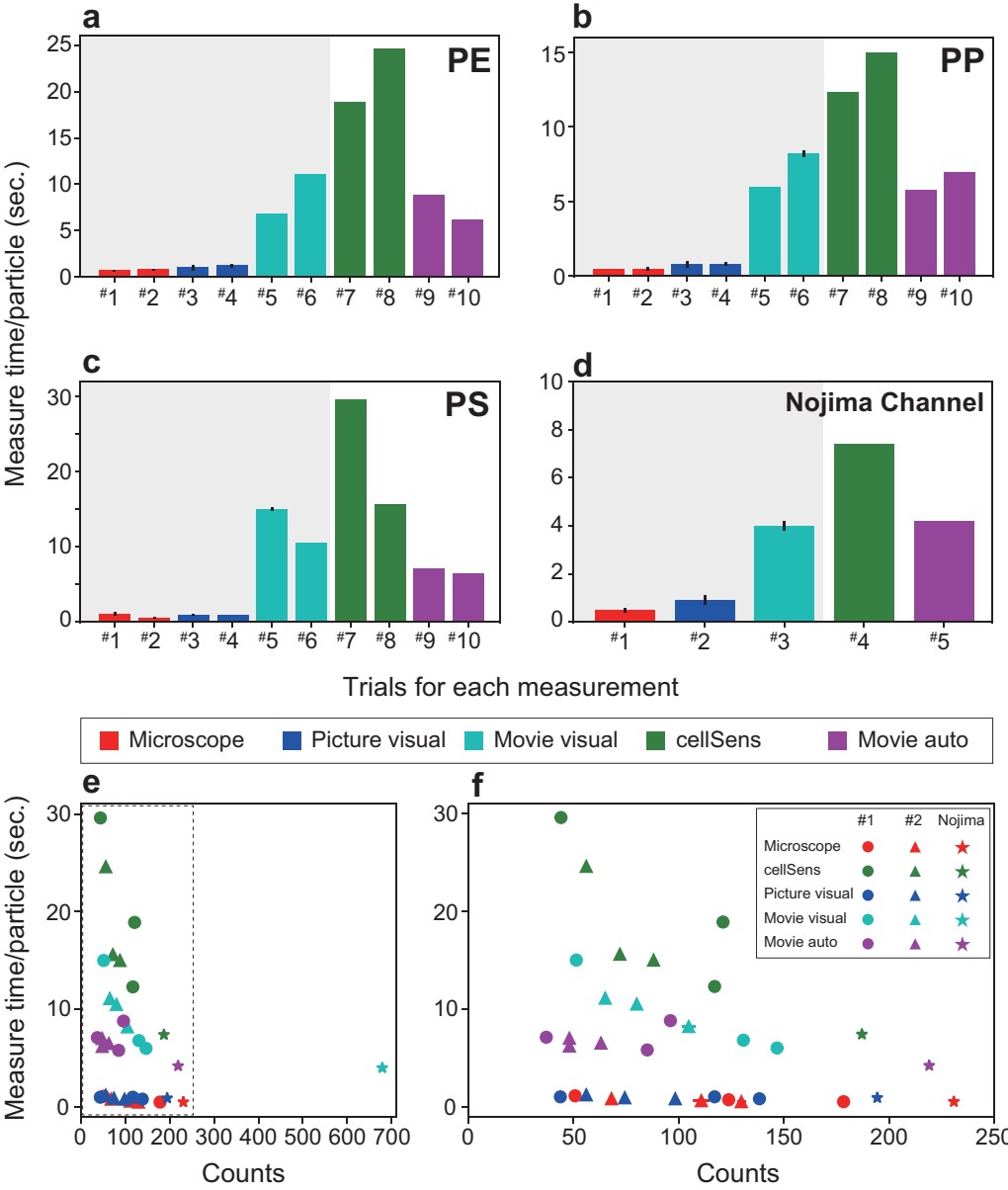

**Figure 4 Comparison of the measurement time per particle and time required for particle detection.** The time required for each material is shown: (A) polyethylene, (B) polypropylene, (C) polystyrene. (D) Measurement time for a sample from Nojima Channel. (E) Time required for particle detection for each measurement method. (F) The enlarged dotted frame in (E). The time taken for one particle in automatic measurement is short and does not take long even if the number of counts increases. As the number of particles increases, the total time required for measurement becomes longer. Therefore, the SAMPdetect method has a relative advantage. Red: visual fluorescence microscopy (Microscope), blue: visual image (Picture visual), light blue: visual video (Movie visual), green: automated count on the software (cellSens), and purple: automatic video (Movie auto). Red, blue, and light blue are particle counts only. Green captures the number of particles plus various parameters for size and area. Purple can acquire particle number and size information. In figures A–D, the horizontal axis shows the trial number and the vertical axis shows the measurement time per particle. Three measurements were made per sample for visual microscopy, image measurement, and video measurement. Error bars indicate standard deviation.

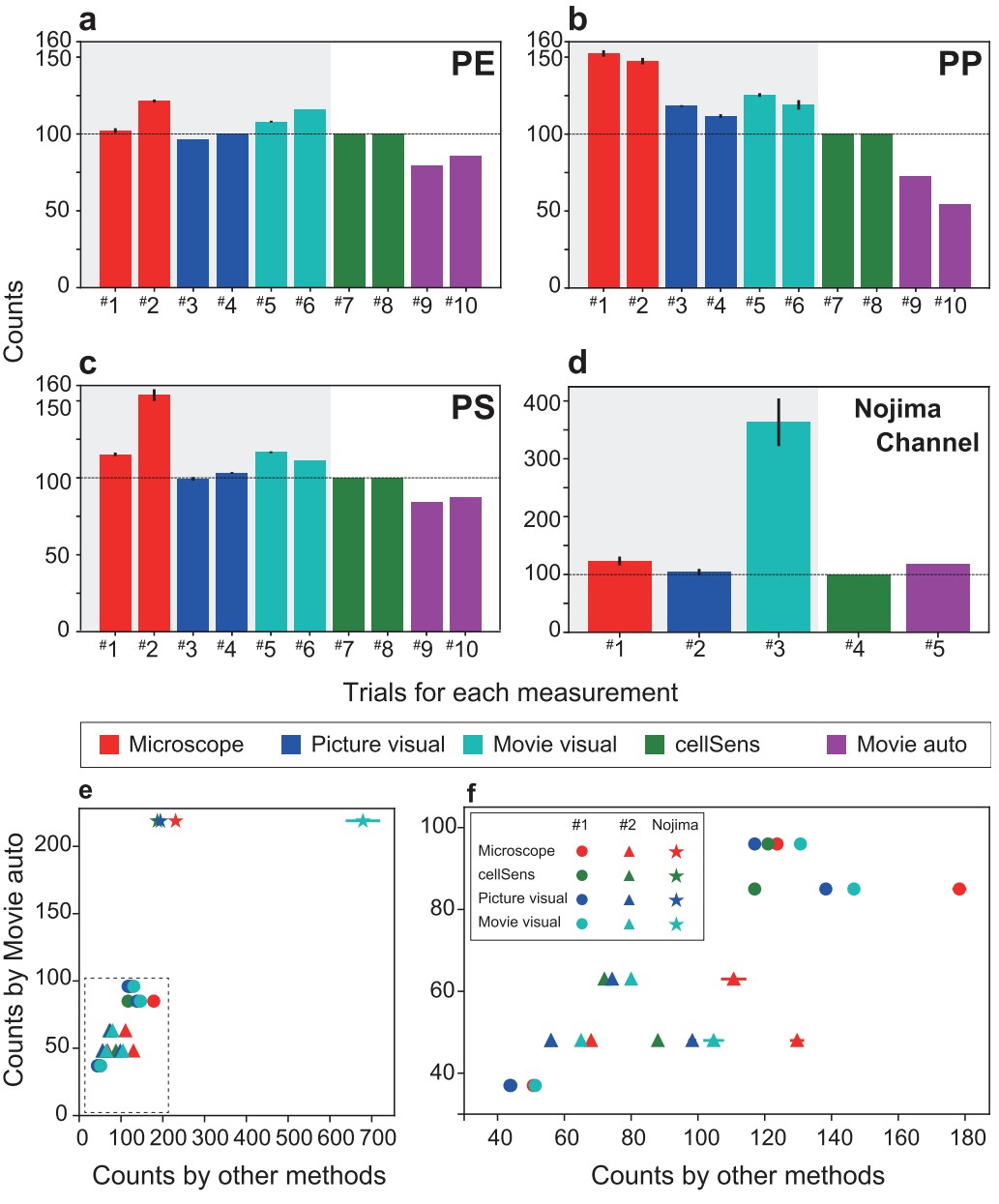

**Figure 5 Comparison of the number of particles detected (detection efficiency) and number of particles detected by each measurement method when the number of particles counted by cellSens is set to 100.** The number of particles counted for each material is shown: (A) polyethylene, (B) poly-propylene, (C) polystyrene, and (D) Nojima Channel samples. (E) The number of particles counted by FlowCell auto-counting compared with the number of particles counted by each measurement method. (F) Enlargement of the dotted frame in (E) showing that FlowCell auto-counting is significantly corre-lated with the previous methods. In contrast, visual observation under the microscope shows over-counting and low correlation. The colors are the same as those shown in Fig. 4. In (A)–(D), the horizontal axis shows the trial number and the vertical axis shows the number of counts. Three measurements were made per sample for visual microscopy, image measurement, and video measurement. Error bars indicate standard deviation.                  

Fenton's reagent was added to the concentrator, which was set in a beaker. Subsequently, an equal volume of 30% $H_2O_2$ was added and the reaction proceeded at 75 °C for 30 min. Care was exercised so that boiling and spilling over did not occur. The chemically treated MPs were subsequently stained with NR (10 μg/mL in hexane) for 1 h in the same concentrator. Then, the NR-stained MPs were poured into the upstream funnel and loaded into the flow cell. This method can be used to remove a large amount of organic matter. The method is particularly useful in deep-sea bottom sediments where the amount of large particulate organic matter is small.

The use of multiple concentrators and beakers or multiple processes increases the risk of MP contamination. For this reason, all treatments were performed using a single concentrator during every step in the processing of each sample. In addition, we conducted the above steps in a laminar airflow cabinet (clean bench) and wore a cotton lab coat to prevent contamination of MPs from the ambient experimental environment (*Wesch et al., 2017*). All reagents used in this experiment were pre-filtered to prevent contamination by MPs. The effectiveness of the flow cell and SAMPdetect method was evaluated using the same method described above to calculate the number of MPs and necessary analysis time.

## Development of an automated detection method

The MPs in the movie captured under the fluorescence stereomicroscope were detected in a defined area at a specific position on the screen (detection window) (Fig. 3A). The detection window was set to any location on the streaming video in the interface, depending on the optimal setting for any given flow rate or numerical particle density of the MPs. We automatically measured the shape (fibers or grains), average particle size (horizontal and vertical feret diameter), and quantity of the NR-dyed MPs passing through the flow cell. The measurement interface was programmed using Python ver. 3.7 (*Van Rossum & Drake, 2009*). To avoid double-counting during particle detection, parameters such as the distance between objects, vertical and horizontal movement range, and their respective ratios were set in the interface.

Stained particles that passed through the movie were marked by squares for fibers and circles for grains, and the size of the particles was measured (Fig. 3B). The Canny method of edge detection was used for image recognition (*Canny, 1986*). A $5 \times 5$ Gaussian filter was applied to reduce the noise in the images with smoothing. Then, the first derivative image was obtained using a Sobel filter, and the contours of the derivative image were narrowed using a non-maximum suppression process. Finally, a hysteresis threshold processing was applied to remove unreliable contours. The edge-detected and recognized particles were then cut out and roughly classified into fibers and grains. The region-of-interest images were then extracted to $100 \times 100$-pixel size and saved as JPEG files. These data were used for training. Because the shape classification in the Python program could not completely discriminate between the two, we performed reclassification by the naked eye to accurately separate them. The fiber and grain data classified by the human eye were used as training data and trained by RAPID Machine Learning and deep learning software (NEC Corporation, Tokyo, Japan). In the case of analysis using natural samples, after size

measurement and image extraction using the Python program, only the extracted JPEG file was analyzed by RAPID for shape identification, without identification by the naked eye.

After detection of MPs, the images of the automatically detected particles were recorded as JPEG format files with unique individual image file names. Simultaneously, the imaged particle name (imaged file name), size (number of pixels), size category, and classification type (grain or fiber) were individually annotated into a cross-referenceable CSV text file. Particle sizes were divided into four categories based on the number of pixels (small, <6 pixels; medium, 6 < 10 pixels; large, 10 < 20 pixels; extra-large, >20 pixels) for descriptive purposes. The size cutoffs corresponded to the pixel ranges, respectively, with one pixel corresponding to 18 μm (small, <108 μm; medium, 108 < 180 μm; large, 180 < 360 μm, and extra-large, >360 μm).

Because particle detection with the script program by Python could detect both fibers and grains, using deep learning is not really necessary. However, the detection confidence was not high; hence, to detect the morphology of the MPs (fibers or grains), we used a combination of a custom-made Python program, and RAPID Machine Learning and deep learning software (NEC Corporation, Tokyo, Japan), a deep learning software package, that can be operated with a graphical user interface. A convolutional neural network classification model was automatically constructed using RAPID based on training data from the particle images retrieved as JPEG files. The JPEG images created with the Python program were used as training data for the machine learning to detect MPs, and the training was repeated up to 200 times. To increase the training data, mirror image processing was used for multiplying the number of training data images. We retrieved 1,513 images of grains and 300 images of fibers, randomly selected 250 images of each of the fibers and grains out of 1,813 images in total, and rotated each image by 90 degrees and used this as training data to determine the type of MPs by RAPID. We evaluated this system using 25 randomly selected images of grains and fibers and calculated the confidence level (%) for the classification of each particle.

## RESULTS AND DISCUSSION

The flow cell-based semi-automated MP particle measurement system developed in this study allowed us to construct a system that efficiently discriminates the number and shape of particles in a short time. As a result, the system is capable of efficiently detecting minute MP particles, which had been a time and labor-limiting factor in MP analysis. This development result enables us to efficiently provide preliminary data on MP distribution patterns while collecting primary data. It will also enable rapid onboard acquisition of primary information during oceanographic surveys, leading to the efficient selection of observation sites.

Using our semi-automated MP detection system, we were able to detect NR-stained MPs in the flow cell and capture the MP video under a fluorescence stereomicroscope (Fig. 3). When the particles flowed through the flow cell, we were able to discriminate, determine the shape and average particle size, and count the number of MPs that passed through the detection frame (Fig. 3B). We also collected the discharged MPs and checked each part of the system for any remaining MPs. The magnification of the objective lens was

factored, and the system was able to observe MPs as small as 50 μm, with the lower limit of detection down to 18 μm for a single pixel.

For each virgin and natural MP sample, a total of seven measurements were performed using each measurement method, detecting 6.8 to 14.2 particles per minute (Supplemental Table). A reduction in detection time was achieved for artificial and natural samples, confirming the improved efficiency of the new system. For different types of samples, the analysis time of the method using the SAMPdetect system (Movie auto) was the shorter (4.2–8.8 s per particle) than cellSens (Figs. 4A–4C, Supplemental Table). Counting under a microscope (Microscope) took 0.5–1.1 s per particle and visual observation with a captured image took 0.8–1.2 s per particle (Figs. 4A–4D), however, size measurements must still be performed separately. Counting and measuring the photographed images in the software (cellSens) took 7.4–29.6 s per particle, while counting by visual observation while playing the movie at twice the speed (Movie visual) took 4.0–15.0 s per particle. In this case, cellSens was able to measure the size. However, in the case of movie visual measurement, only counting was accomplished. This shows that the new system improved the counting and measuring process.

The detection time varied greatly depending on the number of particles and the method of detection used (Figs. 4E, 4F). The detection time per particle decreased as the number of particles increased during software counting, captured image measurements, and visual counting of moving images. This is because the detection time depended on a certain area and length of the movie, regardless of the number of particles. In contrast, visual counting using the microscope, visual counting of the photographed images, and automatic video measurement were able to detect particles at a constant speed regardless of the number of particles. Therefore, we were able to collect more information in a shorter time than with manual counting. Compared with the other four methods used in this study, taking measurements using the proposed system further reduced the time.

Although the detection efficiency of the new SAMPdetect method was slightly lower for artificially created MPs, it was similar for natural samples (Fig. 5). The detection efficiency of the built-in software such as cellSens, which has often been used in various fields of study, was examined based on the method of counting and measuring under a microscope. Microscope and movie visual measurements likely overestimated the overall number of particles (Figs. 5A–5C). In the case of detection by software, a threshold value can be set by staining with various concentrations of NR and staining times in advance. However, naked eye observations can lead to count other dark lights, resulting in overestimation. This may be because visual observation with the naked eye tends to misclassify the unstained MPs due to light scattering. Picture visual measurement (Picture visual) yielded almost the same number of detections as software counting and measurement (cellSens). For the new SAMPdetect method (Movie auto), the overall detection rate was lower. This could be because a few particles were out of the field of view, as the field of view of the movie was 93.3% of the flow cell. When the field of view was set to 100%, the detection efficiencies in trials #1 and #2 were 85.0% and 91.9% for PE, 77.8% and 58.4% for PP, and 90.1% and 93.8% for PS, respectively. The detection efficiency of PP was not high, even when the field
width was considered (Fig. 5B). In contrast, in natural samples, the detection efficiency of the new SAMPdetect method was similar with the other methods, including the software measurement method (Fig. 5D). In addition, the new SAMPdetect method showed an almost one-to-one relationship in the number of measured particles with the other counting methods, except when the particles were counted manually from the movie (Figs. 5E, 5F).

The number of MPs in the natural samples was 5.8 per gram dry weight (5,800 particles per kg dry weight). MPs in Tokyo Bay vary with sediment depth *i.e.*, 1,845 to 5,385 particles per kg dry weight have been reported (*Matsuguma et al., 2017*). The counting values obtained by the new SAMPdetect method for sediments used in this study are consistent with previous studies.

Counting of the naked eye observations under a microscope and movie visual measurement overestimated the number of particles because the threshold of the fluorescence intensity cannot be set as described above. In the movie auto measurement (Movie auto), a slightly larger number of particles were detected even when the field of view width was considered. This is most likely due to the shape of the artificially created MPs. The artificially created MPs are around 300 μm in size, with a relatively high circularity (*i.e.*, more circular) and low aspect ratio (*i.e.*, closer to being circular or square). In contrast, natural MPs are around 150 μm in size, with relatively small circularity (*i.e.*, less circular) and a high aspect ratio (*i.e.*, more elongated or rectangular) (Table 1). These differences in shape likely resulted in the differences in detection efficiency. Additionally, irregularly shaped MP particles move irregularly, even if the velocity is constant. Occasionally duplicate counting of the particles occurred and multiple particles exhibiting different movements that came into contact with each other were counted as a single particle. Therefore, this problem was addressed by adjusting the flow velocity and particle concentration, which assisted in improving the accuracy of image recognition. In any cases, the results of our SAMPdetect method are consistent with the results of existing counting and measurement methods for natural MPs. As with the use of the FlowCam system for meiobenthos analysis, to increase the detection efficiency, measuring the sample and then re-counting the refurbished sample at the output would contribute to error reduction by analyzing the same sample multiple times (*Kitahashi et al., 2018*).

We were able to measure small particles around 100 μm, which are often found in deep-sea sediments (Fig. 6). Although the number of counts somewhat differed, the overall size distribution and number of particles were similar. Artificially produced MPs were sieved to ensure particles around 300 μm were distributed relatively evenly (Figs. 6A–6C). In contrast, natural samples had a large number of particles below 200 μm (Fig. 6D). Similar to the MPs in the natural samples (*Erni-Cassola et al., 2017*), small MPs below 300 μm can be analyzed by micro-FTIR. However, the lower detectable limit with our SAMPdetect method is a few tens of micrometers, depending on the detection machinery. In many cases, such small particles have not been measured and their actual condition has

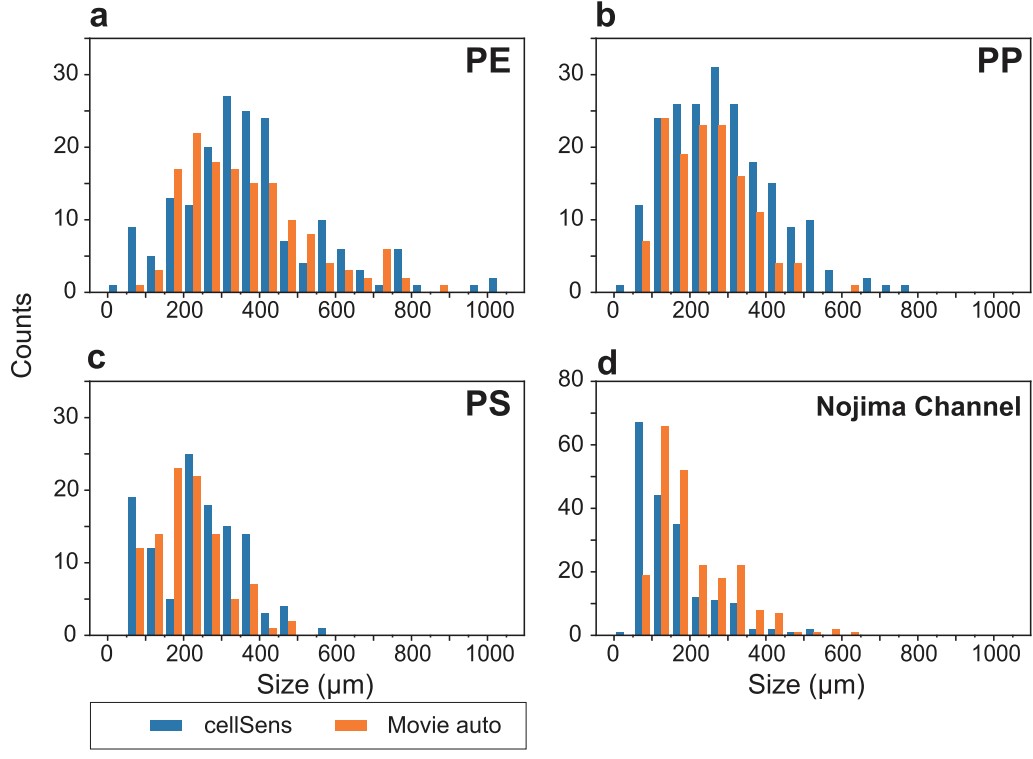

**Figure 6 Particle size distributions for each analysis, showing the results of cellSens measurements (blue) and automatic detection (orange), respectively.** Size distributions of (A) polyethylene (PE), (B) polypropylene (PP), and (C) polystyrene (PS) artificial microplastics (MP) sorted on 100–500 µm sieves. (D) Analytical results of Nile Red-stained MPs in the sediment from Nojima Channel (d).

not been well understood. Although NR staining cannot analyze the material of the plastics, it is an effective tool to reveal their presence and trends (*Erni-Cassola et al., 2017*). Although this study was focused on sediment samples, it can also be applied to suspended MPs in seawater and may be useful for observing grain sizes that have not been analyzed in previous studies.

By using both particulate and fibrous MPs as training data, we were able to discriminate particulate and fibrous MPs with a high confidence level. The confidence level of NR-stained MP particles for three types of particulate MPs plus fibrous MPs flowing in the flow cell is shown in Fig. 7.

In most images, we were able to identify the particles with a confidence of 90% or higher, except for one particle, which had a low confidence level (65%) (Fig. 7A). The low confidence level of the image indicates the visible unevenness of the staining. The fluorescence seen in the outline of the particle was emphasized and misidentified as a fiber, resulting in a lower confidence level (Fig. 7B). Conversely, MPs that were identified as fibers could be identified visually without any misidentification (Fig. 7C). All detected particles had a confidence level of 94% or higher (Fig. 7D). Each of the 25 images were used

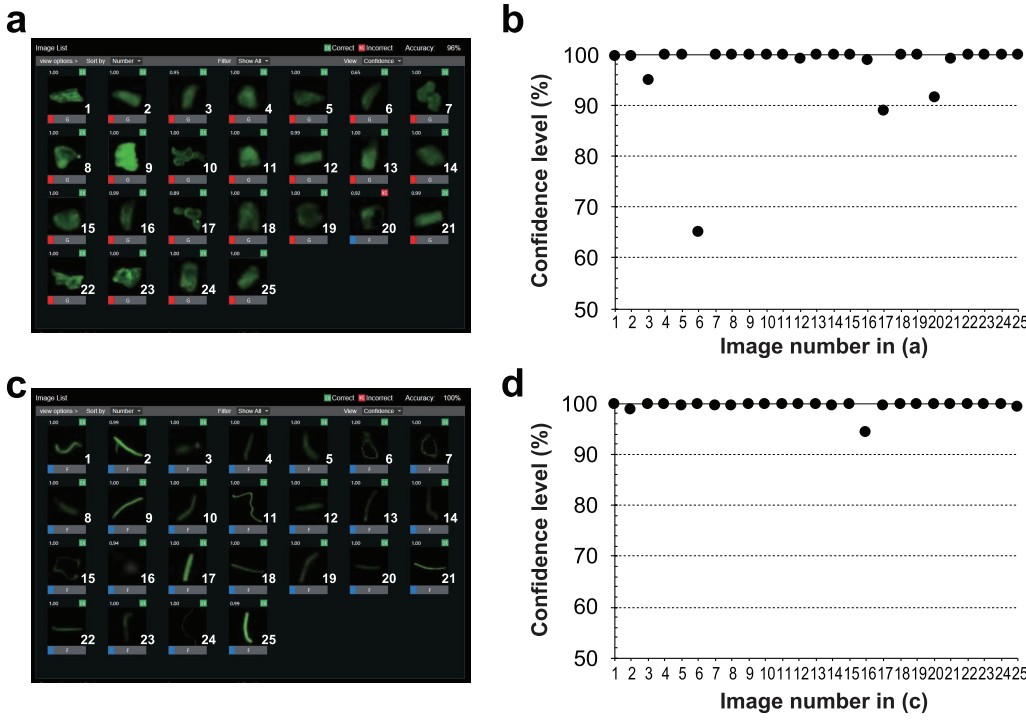

**Figure 7 Classification results and confidence levels for both grains and fibers using the RAPID Machine Learning deep learning software.** Twenty-five evaluated images of (A) grains and (C) fibers, and the raw data of the judged output images. The confidence level (%) for each image of (B) grains and (D) fibers. The horizontal axis indicates the image number and the vertical axis indicates the confidence level (%).                                        

**Table 2 Classification results of the 25 images of both fibers and grains for evaluation.**

|  |  | Number of images for evaluation | |
| --- | --- | --- | --- |
|  |  | Fibers (25) | Grains (25) |
| Number of images evaluated as: | Grains | 0 | 24 |
|  | Fibers | 25 | 1 |
| Correct answers | % | 100.0 | 96.0 |
|  | Average (%) | 98.0 | |

as an evaluation sample. When 90% or higher confidence level was considered as the correct answer, 24 out of the 25 images for particulate MPs had a correct answer—a rate of 96%, while fibers had a correct answer rate of 100% (Table 2). Overall, 98% of the responses were correct. With regard to the learning error, increasing the number of training sessions resulted in an average error rate of 0.0045, indicating that the learning converged with a small error.

Smaller MPs (>10 μm) can be analyzed using FTIR and μFTIR (*Thompson et al., 2004*; *Käppler et al., 2015*), and also it is well known that Raman spectroscopy

(*Van Cauwenberghe et al., 2013*; *Käppler et al., 2015*; *Takahashi et al., 2020*), hyperspectral imaging (*Karlsson et al., 2016*; *Balsi, Esposito & Moroni, 2018*; *Zhu et al., 2020*; *Kitahashi et al., 2021*), and the recently developed laser direct infrared (LDIR) chemical imaging technique (*Bao et al., 2022*; *Cheng et al., 2022*; *Ghanadi et al., 2024*) are effective for quantitative analysis of MPs. For separated MP samples, two techniques are often used to identify the type of MPs. The first method requires manually identifying MPs using FTIR (*Thompson et al., 2004*) or Raman microscopy (*Van Cauwenberghe et al., 2013*). The particles are manually placed individually in the detection area for analysis, making the handling of particles <300 μm with tweezers difficult. The second is the identification of MP particles in a two-dimensional plane by imaging a fixed area, such as using a FTIR imaging microscope. Smaller particles can be detected (generally down to a few 10 μm); however, it takes more than an hour to analyze a surface of 5 × 5 mm area depending on the detection settings. Alternatively, near-infrared hyperspectral imaging can analyze a 50 mm × 50 mm surface in a short period of time (approximately 1 minute (min)) (*Zhu et al., 2020*; *Kitahashi et al., 2021*). All methods have advantages and disadvantages, and one of the problems include the extended time necessary for detecting MPs, especially for μFTIR.

However, applying the methods mentioned above cost high and it is difficult to establish in various sectors. On the other hand, our method, combined with the NR staining method, is simple, and can be used to determine the actual scenario of MP contamination by rapidly analyzing large volumes of samples over a wide area. Especially, our SAMPdetect method is considered to be effective for selecting observation points during a scientific cruise, accumulating primary information, and issuing preliminary values. In addition, this system does not require high costs, making it easily accessible for citizen science. As reviewed comprehensively in *Sinha et al. (2024)*, the role of citizen science goes beyond data collection and has a direct effect toward plastic pollution mitigation and management. Therefore, the participation of many citizens in efforts to understand the actual state of pollution will lead to a better understanding and scientific literacy on all fronts. Our new method does not allow discrimination by material (type of plastic), but the recovered MPs can be analyzed downstream using μ-FTIR, near-infrared wavelength hyperspectral imaging, and Raman spectroscopy, allowing rapid and efficient MP analysis. In fact, coupling NR staining and micro-Raman spectroscopy methods improves the efficiency of MP detection (*Prata et al., 2021*).

Although this study was conducted on sediments, similar techniques have been used for plankton net and other samples. MPs ingested by meiofauna or zooplankton (*Alfonso et al., 2023*), rather than occurring freely in the sediment or water, unfortunately cannot be separately assessed. However, particles smaller than a few tens of micrometers in size, which are difficult to analyze by FTIR microscopy, can be analyzed by changing the magnification of the microscope. In this study, a fluorescent chemical reagent was used to stain MPs; however, recognition of the polymer type of plastic could not be performed. If various types of plastics can be individually labeled, such as with peptides

(*e.g.*, polystyrene-binding peptides, *Kumada et al., 2010*), with various fluorescence techniques (*e.g.*, Q-dot), qualitative analysis can also be performed using a fluorescence microscope.

### Limitations

Development of the SAMPdetect method is a pilot study and can still be improved in some aspects. In fact, the same particles may be counted again because of the irregular movements of the irregularly shaped MPs, even if the flow velocity is the same. In addition, multiple particles with different movements may be counted as a single particle when they come into contact with each other. In some cases, particulate MPs can be misidentified and classified as fibers depending on the fluorescence intensity and staining conditions, even though the staining duration and concentration of NR were fixed. The problem can be solved by adjusting the flow rate and the concentration of MPs, dyeing method, and increasing the amount of training data. As shown in this study, determining the detection efficiency and accuracy rates and the reliability of the data from this system for a natural sample using reference samples, such as artificial MPs, is necessary. Although NR stains organisms and the other organic matter in marine samples, it may lead to false detections. False positives can be reduced by removing organic matter or carbonates with digestion, such as Fenton's reagent (*Tagg et al., 2017*) and hydrochloric acid before NR staining. Recently staining methods with Safranine T or fluorescein isophosphate (*Lv et al., 2019*), and coumarin 6 (*Chen et al., 2023*) have been applied to MP detection, and identify MPs with the combination of filters and machine learning (*Meyers et al., 2022*), and when combined with our new SAMPdetect method, more accurate MP detection might be expected.

## CONCLUSIONS

Our SAMPdetect method can continuously and automatically measure the number and size of NR-stained MPs, and classify them as particulate or fibrous. Using this classification of particulate and fibrous MPs as training data, this system can discriminate between them and automatically measure 6.8–14.2 particles per minute. When the automatic detection system was combined with RAPID software, the correct answer rate for the detection of artificial MPs was 96% for particles and 100% for fibers (98% on average). As both the number and size of particles can be measured simultaneously, the time required is reduced compared to the previous methods. Although the number of particles measured per unit of sediment was comparable to the previous studies in natural samples, the detection efficiency was reduced in a few cases for artificial MPs. This could be attributed to the shape of the particles. Although our system cannot discriminate between the polymer type of MPs, we can obtain primary information to acquire an overall picture of the actual condition of MP pollution. This system costs inexpensive to compare with the other systems and can be used in various applications and sectors, including a scene at citizen science.

## ACKNOWLEDGEMENTS

We thank Rie Matsui of JAMSTEC for her help in sediment preparation and Daisuke Kojima and Junpei Murasato of Pearl Giken Co., Ltd. for their help with flow cell fabrication.

### Funding

This research is funded by the Environmental Research and Technology Development Fund (SII-2, JPMEERF18S20211) of the Environmental Restoration and Conservation Agency of Japan (ERCA) and the Danish National Research Foundation through the Danish Center for Hadal Research (grant number DNRF145). The funders had no role in study design, data collection and analysis, decision to publish, or preparation of the manuscript.

### Grant Disclosures

The following grant information was disclosed by the authors:
Environmental Research and Technology Development Fund: SII-2, JPMEERF18S20211.
Environmental Restoration and Conservation Agency of Japan (ERCA).
Danish National Research Foundation: DNRF145.

### Competing Interests

The authors declare that they have no competing interests. Yosuke Taira and Hitoshi Saito are employed by NEC Corporation and Tomo Kitahashi is employed by KANSO Technos Co., Ltd.

### Author Contributions

- Masashi Tsuchiya conceived and designed the experiments, performed the experiments, analyzed the data, prepared figures and/or tables, authored or reviewed drafts of the article, and approved the final draft.
- Tomo Kitahashi conceived and designed the experiments, performed the experiments, analyzed the data, prepared figures and/or tables, authored or reviewed drafts of the article, and approved the final draft.
- Yosuke Taira conceived and designed the experiments, authored or reviewed drafts of the article, and approved the final draft.
- Hitoshi Saito conceived and designed the experiments, authored or reviewed drafts of the article, and approved the final draft.
- Kazumasa Oguri conceived and designed the experiments, authored or reviewed drafts of the article, and approved the final draft.
- Ryota Nakajima conceived and designed the experiments, authored or reviewed drafts of the article, and approved the final draft.
- Dhugal J. Lindsay conceived and designed the experiments, authored or reviewed drafts of the article, and approved the final draft.
- Katsunori Fujikura conceived and designed the experiments, authored or reviewed drafts of the article, and approved the final draft.

## Data Availability

The python script is available at GitHub and Zenodo:

- https://github.com/tkitahashi/SAMPdetect.

- tkitahashi. (2024). tkitahashi/SAMPdetect: SAMPdetect (python). Zenodo. https://doi.org/10.5281/zenodo.13925395.

The original images and movies are available at FigShare:

- TSUCHIYA, MASASHI; Kitahashi, Tomo; Taira, Yosuke; Saito, Hitoshi; Oguri, Kazumasa; Nakajima, Ryota; et al. (2024). Raw data of 1. Microscope, 2. Picture visual and 4. cellSens measurements for "Rapid detection and quantification of Nile Red-stained microplastic particles in sediment samples". figshare. Dataset. https://doi.org/10.6084/m9.figshare.27201732.v2.

- TSUCHIYA, MASASHI; Kitahashi, Tomo; Taira, Yosuke; Saito, Hitoshi; Oguri, Kazumasa; Nakajima, Ryota; et al. (2024). Raw data (input movie) for 3. Movie visual and 5. SAMPdetect for "Rapid detection and quantification of Nile Red-stained microplastic particles in sediment samples". figshare. Media. https://doi.org/10.6084/m9.figshare.27209871.v1.

- TSUCHIYA, MASASHI; Kitahashi, Tomo; Taira, Yosuke; Saito, Hitoshi; Oguri, Kazumasa; Nakajima, Ryota; et al. (2024). Raw data of output files of 5. SAMPdetect for "Rapid detection and quantification of Nile Red-stained microplastic particles in sediment samples". figshare. Dataset. https://doi.org/10.6084/m9.figshare.27209898.v1.

## Supplemental Information

Supplemental information for this article can be found online at http://dx.doi.org/10.7717/peerj.19196#supplemental-information.

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
