# Peer review of "Rapid detection and quantification of Nile Red-stained microplastic particles in sediment samples"

_PeerJ, doi:10.7717/peerj.19196_

## Round 0.1 · original submission · Major Revisions

The SAMPdetect method established in this study provides a promising solution to the current challenges of time-consuming and labor-intensive microplastic detection in environmental samples. By leveraging machine learning for model training, the system design is both comprehensive and rigorous. Additionally, the method incorporates multiple analytical approaches for comparison. However, major revisions are needed as per the comments of the reviewers.

Reviewer 1 ·

Basic reporting

The SAMPdetect method established in this study provides a promising solution to the current challenges of time-consuming and labor-intensive microplastic detection in environmental samples. By leveraging machine learning for model training, the system design is both comprehensive and rigorous. Additionally, the method incorporates multiple analytical approaches for comparison. It demonstrates high resolution, with a detection limit as low as 18 µm. However, there are some formatting issues in the manuscript that need attention.
1. Why do the bar charts in Figures 4 and 5 have error bars in some cases but not in others? The figure legend includes annotations for the number of repetitions and error bars, but this is only mentioned for Figure 5, not Figure 4.
2. Line 268: The expression "<108 µm, 108<180 µm, 180<360 µm and >360 µm" needs to be revised for clarity.

Experimental design

One limitation of the NR technique is the difficulty in distinguishing between natural particles and plastic particles. Although the authors mention the use of digestion methods, environmental samples often contain complex organic matter, including plant and biological tissues, which are difficult to digest. How can the authors ensure that all natural materials are fully digested during the detection process?

Validity of the findings

1. Although the study compares the time consumption of FTIR and Raman detection, the proposed method does not include identification capabilities and can only detect three types of plastics (PP, PS, and PE). Given the increasing variety of plastic types, this may not be considered highly advanced in the current context. Could the authors discuss whether it is possible to expand the range of detectable plastic types in the future? Alternatively, as mentioned by the authors, can combining this method with other detection technologies overcome the limitations of NR in detecting a wider range of plastics? This may require some discussion of the practical applications of the method to highlight its applicability.
2. The NR staining technique is a relatively classical method, and there are existing studies that combine Nile Red staining with machine learning for microplastic detection, such as:
https://doi.org/10.1007/s11356-024-35289-0
https://doi.org/10.1016/j.marpolbul.2024.116804

What are the advantages of the method proposed in this study? It would be helpful to add this in the discussion and highlight the innovative aspects of your approach within the manuscript.

Reviewer 2 ·

Basic reporting

no comment

Experimental design

no comment

Validity of the findings

no comment

Additional comments

The research examined the rapid identification and measurement of Nile Red stained microplastic particles in sediment samples. The study is really intriguing about its topic matter. The extraction of microplastics (MPs), their microscopic analysis, preliminary identification, and subsequent determination of polymer types in MP investigations are lengthy and labor-intensive operations. In this regard, it is evident that the strategy outlined in this work effectively conserves time in the identification and characterization of MPs. The system's overall evaluation reveals that the extracted MPs can be identified at a specific velocity and without necessitating direct eye examination. But, Nile red is not appropriate for detecting all forms of polymers. So, There exists a likelihood that substantial quantities of MPs may remain undetected. Nile red can pose significant challenges for dyeing fiber particles. In addition, The system may encounter difficulties in recognizing overlapping particles. Does this system autonomously modify the depth? The necessary visual clarity may not always be attained. Furthermore, Operating within a limited range of MP lengths also implies that further particles are excluded from consideration. The computation of these particles will necessitate an additional labor. Ultimately, as MP falls within the 1-5000 µm particle range, this spectrum is typically sought for in research. It is crucial to analyze each type of polymer individually. Indeed, the numerical value is significant; nevertheless, the type of polymer is also a crucial aspect. The necessity of a labor force for this raises doubts about the efficiency of the system designed to conserve time. Moreover, manufacturers of microscopes supply software that automatically analyzes them using a fluorescence microscope through a mapping approach. What advantages might this system possess compared to the one I mentioned? Time efficiency is noted; however, Raman systems may also autonomously scan and capture images of specific regions (filter surface) and detect polymer kinds. Furthermore, the recently employed LDIR Chemical Imaging Spectroscopy equipment can accomplish this with greater precision. I believe it is necessary to reassess the publishing concerning all the aforementioned aspects.

Reviewer 3 ·

Basic reporting

This paper proposes a semi-automated microplastic detection and analysis method from sediment samples. They used fluorescence to distinguish and to detect the microplastic from a liquid samples after performing a density separation.
Lines 40-47 & 101-104: Please rephrase this sentence. The mentioned MP sizes are contradictory.
Lines 99-107: The objective is not clearly outlined.
In order to showcase the citizen science approach of microplastic detection, it is recommended to discuss this Introduction section as well using the latest references (for example, https://doi.org/10.1016/j.scitotenv.2024.176621).

Experimental design

Apropriate

Validity of the findings

validated

Additional comments

none

---

## Round 0.2 · accepted · Accept

This revised version is suitable for publication in PeerJ.

Reviewer 1 ·

Basic reporting

The authors have thoroughly addressed the reviewers' concerns through substantial revisions to their manuscript. I therefore recommend acceptance of the revised manuscript for publication.

Experimental design

no comment

Validity of the findings

no comment

Reviewer 3 ·

Basic reporting

The revised version has improved and incorporated all the reviewer comments. It may be acceptable now.

Experimental design

good

Validity of the findings

reliable

Additional comments

none